# Analysis of Various Facial Expressions of Horses as a Welfare Indicator Using Deep Learning

**DOI:** 10.3390/vetsci10040283

**Published:** 2023-04-10

**Authors:** Su Min Kim, Gil Jae Cho

**Affiliations:** College of Veterinary Medicine, Kyungpook National University, Daegu 41566, Republic of Korea; liebe_sm@naver.com

**Keywords:** automatic recognition, deep learning, equine welfare, facial expression, horse, pain, profile

## Abstract

**Simple Summary:**

Pain assessment in animals depends on the observer’s ability to locate and quantify the pain based on perceptible behavior and physiological patterns. It is currently well established in veterinary medicine that pain trigge rs behavioral changes in animals, and monitoring these changes is important in the assessment of pain and evaluation of the welfare state of an animal. Recently, several studies have been conducted in horses to evaluate their pain based on their expression. However, there was no study that measured the level of pain, whether acute pain and chronic pain could be distinguished, or if other conditions could be mistaken for pain. Thus, studies on pain identification based on various facial expressions that can be misjudged are lacking. In this study, a horse facial expression recognition model was developed to automatically analyze these expressions using deep learning. We captured not only pain expressions but also comfort, tension, and excitation expressions as images by classifying them into four labels: resting horses (RH), horses with pain (HP), horses immediately after exercise (HE), and horseshoeing horses (HH). Furthermore, this study classifies horses’ expressions and presents more objective indicators for animal welfare by analyzing their pain and various expressions.

**Abstract:**

This study aimed to prove that deep learning can be effectively used for identifying various equine facial expressions as welfare indicators. In this study, a total of 749 horses (healthy: 586 and experiencing pain: 163) were investigated. Moreover, a model for recognizing facial expressions based on images and their classification into four categories, i.e., resting horses (RH), horses with pain (HP), horses immediately after exercise (HE), and horseshoeing horses (HH), was developed. The normalization of equine facial posture revealed that the profile (99.45%) had higher accuracy than the front (97.59%). The eyes–nose–ears detection model achieved an accuracy of 98.75% in training, 81.44% in validation, and 88.1% in testing, with an average accuracy of 89.43%. Overall, the average classification accuracy was high; however, the accuracy of pain classification was low. These results imply that various facial expressions in addition to pain may exist in horses depending on the situation, degree of pain, and type of pain experienced by horses. Furthermore, automatic pain and stress recognition would greatly enhance the identification of pain and other emotional states, thereby improving the quality of equine welfare.

## 1. Introduction

Facial expressions in animals were first studied by Charles Darwin [1] while investigating the expression of emotions in humans and animals. Since then, facial expressions in animals have been studied as tools for pain assessment and have gained popularity as such [2,3].

The fact that pain induces behavioral changes in animals is well established in veterinary medicine, and pain monitoring has become important for the estimation of pain and evaluation of animal welfare status [4].

In general, animal behavioral and activity pattern studies are conducted manually or semimanually via human observation, which is subjective and has its limitations [5,6,7]. Modern horse pain scales utilize a variety of tools based on behavioral and physiological parameters, and it has been demonstrated that behavioral parameters are more specific for pain than physiological parameters [8,9,10,11,12,13]. It is generally accepted that a single physiological or biochemical parameter for horse pain is not practical [14,15]. In particular, studies on physiological parameters, such as heart rate and cortisol blood levels, have demonstrated that these parameters are closely related to pain [8]. However, several studies have concluded that in general, physiological parameters are weakly associated with pain, whereas behavioral parameters are often more easily assessed and considered more pain-specific [9,10,11,12,16]. Furthermore, although physiological parameters may be valid as pain indicators in highly controlled environments, most are invasive and require stressful blood collection or animal restraint [17].

Unlike humans, who can express pain through verbal communication, animals depend on the observer’s ability to determine and quantify their pain based on perceptible behavioral and physiological patterns [18]. Animals cannot express pain, and prey animals have evolved to hide pain in the wild [11,19]. In particular, prey animals, such as horses, exhibit less-pronounced pain behavior in the presence of veterinarians or unfamiliar human observers [20,21]. A recent study revealed that the feeling of discomfort is often underestimated because postoperative discomfort behavior becomes less apparent when caretakers communicate with horses [22]. Thus, pain recognition in animals by humans is not only difficult but also requires considerable training and experience.

Horses’ facial expressions are often dynamically complex cues that can rapidly change in response to various environmental stimuli and internal emotional states [23]. In addition to pain, the facial expressions of ridden horses can reflect responses to a variety of stimuli, including responses to signals from the rider, pressure from the saddle and tack [24,25], and interactions with the surroundings. Therefore, it is necessary to consider this complexity while designing studies to evaluate the expression of horses [26]. Despite the importance of this topic, there is no “gold standard” for identifying and managing various equine expressions aside from pain.

In recent decades, some tools have been developed primarily for the diagnosis and evaluation of horse pain in certain clinical situations [9,13,27,28]. Furthermore, manually scored pain assessment runs are repeatedly performed by an observer, particularly with an increased risk of noncorrespondance [29]. Despite the validity gained for the distinction between healthy and pain-experiencing horses, the observer’s ability to recognize the normal behavior of a horse can have inconsistent results [29]. Thus, there is an increased risk of underestimating pain or misinterpreting behavior in the context of assessment of well-being and quality of life [29]. Therefore, the use of machine learning for pain recognition can not only increase the likelihood of monitoring and changing the perception of pain but can also become an important and essential interactive educational tool to learn from expert experience and novice training [26].

With the development of deep learning technology, novel image recognition technologies are rapidly developing [30]. Deep learning is a technique for learning a deep neural network structure composed of multiple layers using a large amount of data [30]. It is known that the ability to learn nonlinear hierarchical features is similar to human cognitive mechanisms [30]. The development of deep learning techniques does not require feature engineering prior to classification or regression and can be monitored in a noninvasive manner [30].

Recent advances in computer vision, image processing, and deep learning have enhanced behavioral science research [31,32,33]. Furthermore, advances in veterinary medicine in recent years have been increasingly directed toward the replacement of human subjective and often unreproducible behavioral assessments, e.g., those regarding lameness in horses, using computer technology [34,35]. Recently, research on activity measurement and walk analysis using accelerometer data and convolutional neural networks (CNN) has been conducted [34,35]. The results of these studies indicate that artificial intelligence can be used to continuously monitor horse behavior [29].

This study has the following hypotheses: (1) horses have various facial expressions, such as resting horses (RH), horses with pain (HP), horses immediately after exercise (HE), and horseshoeing horses (HH) and (2) these facial expressions can be evaluated by developing an image-based automation system that distinguishes and recognizes them. Therefore, the purpose of this study is to present more objective indicators for animal welfare by analyzing various facial expressions of horses.

## 2. Materials and Methods

### 2.1. Animals

A total of 749 horses (586 healthy horses and 163 horses with pain) were included in this study; these horses were raised on the racing courses of the Korea Racing Authority (KRA) and public horse-riding clubs in July 2020 and August 2021. The horses rested in their individual stables for 24 h, except during exercising, riding, or horseshoeing. They were provided water freely and high-quality feed and hay thrice a day.

To observe various expressions according to the purpose of our study, we classified the horses and captured their expressions as follows:RH (*n* = 210): Horses that rested in stables. We confirmed from owners that the horses were not exposed to any external stimuli, exercise, or horseshoeing and from veterinarians that the horses had no disease 3 h before filming.HP (*n* = 163): Most horses with pain due to complications, such as acute colic, radial nerve paralysis, laminitis, and joint tumors, were transported to an equine veterinary hospital of the KRA. These horses were diagnosed by veterinarians and underwent surgery, treatment, and hospitalization (N = 136/163). Fattened horses, primarily Jeju horses in South Korea, raised on fattening farms were also included (N = 8/163). Their feces and urine were not removed from the stables; all horse hooves were turned, and the horses could not walk properly. Furthermore, the images of horses with pain used in the previously published papers were extracted and included in this study (N = 19/163).HE (*n* = 156): Horses that were well trained and exercised every day were included in this study after the veterinarians confirmed that they were free from edema or lameness. Horse exercises included walking, trotting, cantering, and horse riding for 50 min and treadmill for 40 min. Filming was performed immediately after the exercise and dismantling the bridle.HH (*n* = 210): These are horses that had to have farriery treatment by farriers. On the day of horseshoeing, the horses did not exercise and rested in their individual stables. Images were taken with the horse fixed after the halter was fastened with both lead straps in the stable corridor during farriery treatment. Horses without visible lameness were included prior to horseshoeing.

### 2.2. Image Collection

All horses included in this study had no visible scars or cuts on their face, and the owners of the horses agreed to image collection. The researchers approached the horse, gently stroked it for familiarity, and attempted to minimize irritation and stress. After fixing the horse with a halter and lead strap to photograph its face, it was filmed from multiple angles (left, between left and front, front, between front and right, and right; three pictures each, with a total of 15 pictures per horse) (Figure 1). Images were captured without any noise using the built-in camera of a cellular phone. The equine faces were calibrated with a distance of approximately 50 cm from camera, and images of horses with blinking eyes or raising heads were excluded.

## 3. Proposed Methodology

### 3.1. Method Overview

We tested a mechanism to recognize facial expressions of horses based on the captured images. We trained the model on a subset of that dataset (*n* = 339), validated it on a validation set (*n* = 137), and reported the accuracy over the test set (*n* = 273).

Before describing the details of the model’s performance on different datasets, we briefly discuss our training procedure. To normalize the equine facial posture, a profile with a high recognition rate was selected among the horse images taken from various angles using the residual network (ResNet) 50 model. For the objective analysis of the horse’s face, three keypoints, namely, eyes, nose, and ears, were selected on the basis of an equine facial action coding system (EquiFACS) [36] and grimace face scale (HGS) [37]. Features were extracted by masking the other parts with black. Each image sample uses one of four labels (RH, HP, HE and HH); the framework of the model is presented in Figure 2. Data augmentation was used for the images in the training set to train the model on a larger number of images and on invariance in small transformations [38]. Flipping, small rotation, and small distortion were used to augment the data [39]. A model based on PyTorch was constructed (ver. 1.7.1., Facebook AI Research, Menlo Park, CA, USA). PyTorch is an open-source machine learning framework that accelerates the path from research prototyping to production deployment.

### 3.2. Normalization of Facial Posture

Facial expression is influenced by lighting, head posture, environment, and other factors, which cause significant differences in the performance of different images in the network [40]. Dataset preprocessing can effectively eliminate redundant information and reduce data noise, thereby improving the effectiveness of facial expression feature extraction [40]. In a natural environment, the detected equine faces may have different angles according to the head posture, and we need a preprocessing step to minimize the effect. In the present study, we developed a “profile classification model” that automatically extracts a profile image, which is easier for expression recognition, among images taken from multiple angles of a horse’s face.

To improve the learning performance and speed, the ResNet 50 model was used after training using the ImageNet-based pretrained model, which has a large amount of image data [41]. The ResNet 50 model was proposed by He et al. [42], and it has the advantage of interpreting the problem of training deep neural networks using the convolutional features of the ResNet 50 model trained through massive ImageNet data.

Figure 3A presents the network structure of each layer of the ResNet 50 model in detail. Figure 3B shows that it can be realized using a feedforward neural network through a shortcut connection.

### 3.3. Equine Facial Keypoint Detection for Classification

In this study, we used horse EquiFACS [36] and HGS [37] to select the keypoints of the eyes, nose, and ears that most commonly affect the horse’s pain expression. We also developed an “eyes–nose–ears detection model” that extracts them from the horse’s face. After the detection model recognizes the eyes, nose, and ears, it automatically crops the horse’s face area and masks the rest in black. In this process, if the images in the background are complex, it may be difficult to recognize the eyes, nose, and ears; therefore, it was rotated and manually labeled. The masked images were resized to 224 × 224 pixels and applied to the classification model to derive the result.

We used the EfficientDet model, the latest model that improves both the speed and accuracy of object detection. This model uses a weighted bidirectional feature pyramid network, which allows for easy and fast multiscale feature fusion [43]. Furthermore, by applying a complex scaling method that uniformly scales resolution, depth, and width for all backbones, feature networks, and box/class prediction networks at the same time, it exhibits much higher efficiency than the existing techniques in a wide range of resource constraints [43]. Figure 4 illustrates the structure of EfficientDet; EfficientDet D4, which is known to be an appropriate level among D1–D7, was used.

### 3.4. Class Activation Mapping

We used a network architecture similar to Network in Network [44] and GoogLeNet [45], which mainly consist of convolutional layers, and just before the final output layer (softmax, in the case of categorization), we performed global mean pooling on the convolutional feature maps and used those as features for a fully connected layer that produces the desired output (categorical or otherwise) [46]. Owing to this simple connectivity structure, we can identify the importance of the image regions by projecting back the weights of the output layer onto the convolutional feature maps, a technique known as class activation mapping [46].

As shown in Figure 5, global mean pooling outputs the spatial average of each unit’s feature map at the last convolutional layer. We computed a weighted sum of the feature maps of the last convolutional layer to obtain our class activation maps (CAMs) [46]. Referring to a study by Zhou et al. [46], we describe this more formally below for the case of softmax.

For a given image, let $f_{k}\left(x,y\right)$ represent the activation of the unit k in the last convolutional layer at the spatial location (x,y). Then, for unit k, the result of performing global average pooling $F^{k}$ is $\underset{x,y}{\sum }f_{k}\left(x,y\right)$. Thus, for a given class c, the input to the softmax $S_{c}$ is $\underset{k}{\sum }{𝓌}_{k}^{c}F_{k}$, where ${𝓌}_{k}^{c}$ is the weight corresponding to class c for unit k. Essentially, ${𝓌}_{k}^{c}$ indicates the importance of $F_{k}$ for class c. Finally, the output of the softmax for class c $P_{c}$ is given by $\frac{exp\left(S_{c}\right)}{\sum_{c}exp\left(S_{c}\right)}$. Here, we ignore the bias term; we explicitly set the input bias of the softmax to zero, as it has little to no impact on the classification performance.

By plugging $F_{k}=$ $\underset{x,y}{\sum }f_{k}\left(x,y\right)$ into the class score $S_{c}$, we obtained the following equation:(1)$$S_{c}=\sum_{k}{𝓌}_{k}^{c}\sum_{x,y}f_{k}\left(x,y\right)\;=\sum_{x,y}\sum_{k}{𝓌}_{k}^{c}f_{k}\left(x,y\right)$$

We define M_c as the CAM for class c, where each spatial element is given by
(2)$$M_{c}\left(x,y\right)=\sum_{k}{𝓌}_{k}^{c}f_{k}\left(x,y\right)$$

Therefore, $S_{c}$ = $\underset{x,y}{\sum }M_{c}\left(x,y\right)$, and hence, $M_{c}\left(x,y\right)$ directly indicates the importance of the activation at the spatial grid (x,y), leading to the classification of an image to class c.

Intuitively, based on previous studies [46,47], we expect each unit to be activated by some visual pattern within its receptive field. The CAM is simply a weighted linear sum of the presence of these visual patterns at different spatial locations [46]. By simply upsampling the CAM to the size of the input image, we can identify the image regions that are most relevant to a particular category [46].

### 3.5. Model Convergence

The classification accuracy of the developed model was evaluated using four labels (RH, HP, HE and HH). Accuracy is an indicator for evaluating classification models. The classification model also returns true and false, so the results are presented as true and false, and the case divided into the 4 × 4 matrix is presented in Table 1.

The scale adopted in this study is the accuracy commonly used to evaluate categorization techniques, and the accuracy ($\alpha_{i}$) is defined as follows:(3)$$\alpha_{i}=\frac{TP_{i}+TN_{i}}{TP_{i}+TN_{i}+FP_{i}+FN_{i}}$$

Cross-entropy (*CE*) loss is actually the only loss we will discuss here. The *CE* loss is defined as follows:(4)$$CE=-\sum_{i}^{C}t_{i}\log\left(s_{i}\right)$$
where $t_{i}$ is the ground truth (correct answer), and $s_{i}$ is the ith element of the score vector, which is the output of the last layer of the CNN for each class i. To fit the calculation range between (0, 1), the score is computed with the *CE* loss, often combined with the sigmoid activation function.

## 4. Results

### 4.1. Normalization of Facial Posture Results

Table 2 presents the normalization results of equine facial posture by the profile classification model. The profile presented higher accuracy than the front. Furthermore, this model accepts images taken from a profile between 45° and 90° of the sagittal plane of the horse.

### 4.2. Visualization of Class Activation Mapping

In our study, we provide a simple approach to visualize important areas for classifying the facial expressions of horses into four layers. The output image shows the weight distribution after visualization, where red and blue indicate high and low area weights, respectively. As presented in Figure 6, the eyes, nose, and ears are the important parts for concentrated expression, but the highest weight distribution is concentrated on the eyes.

### 4.3. Equine Facial Keypoint Detection Classification Performance

In our study, of the 749 samples, 339 were used for training, 137 were used for validation, and 273 were used for testing. The classification accuracy of the trained model obtained in the experiment is presented in Table 3. The eyes–nose–ears detection model achieved an accuracy of 98.75% in training, 81.44% in validation, and 88.1% in testing, with an average accuracy of 89.43%.

Overall, the average classification accuracy of the four labels was high, but the accuracy of pain classification was the lowest. This was analyzed in more detail in horses experiencing pain, as their expression may be different depending on the degree of pain. In the training and validation processes, the images of horses that suffered extreme pain due to acute colic, laminitis, radial nerve paralysis, or joint tumor were used. In the testing process, the scope was widened to include papers, Google search results, post-surgery, laminitis, fattened horses, and severe lameness. The results are presented in Table 4. Papers, Google search results, surgery results, and laminitis results suggested that there was severe pain, as confirmed in the training and validation processes; however, the results of fattened horses and horses with severe lameness were surprising. The breeding conditions of the fattened horses that lived in private stables were unhygienic because night soil was not removed. Moreover, the hooves were not trimmed for a long time, which resulted in chronic damage to the hooves and fetlocks. Furthermore, horses with severe lameness were resting in their own stables when the researchers were filming; thus, they appeared not to be feeling pain as they were not using their legs. This implies that horses exhibit different pain expressions depending on the situation, degree of pain, and type of pain they are experiencing.

### 4.4. Model Convergence

Here, we present the model classification accuracy and loss on the validation set during the training process to improve our understanding of the model convergence speed. The model performance on the validation set is illustrated in Figure 7. As shown in the figure, the general trend in verification accuracy increases over time, but some vibrations exist at some section. These vibrations can be avoided by selecting smaller learning rates, but convergence rates can be slowed when training models. Finally, it is worth mentioning that the model with the highest verification accuracy is used to indicate a test error.

Accordingly, we plotted the measure of time or progress on the x-axis and the measure of error or performance on the y-axis. We used these charts to monitor the evolution of our model during learning so that we could diagnose problems and optimize the prediction performance.

## 5. Discussion

In recent years, public interest in horse welfare has significantly increased, as the waste and breakdown of equestrian sports becomes known [48,49]. Studies on the equestrian field have demonstrated that repetitive strain injuries are likely a precursor to these events. Therefore, early diagnosis is very important [50]. Welfare issues are relevant for stakeholders, horse owners, horse managers, and veterinarians [17].

Horse pain assessment has several important problems. For example, compared with the immense amount of data on human faces, information for examining animal faces is extremely limited [18]. In fact, it is not only difficult to collect human expression datasets, such as those that reveal human faces in full frontal view of the camera, but they also have limited generalization to natural settings, where a face is likely to move in and out of camera view [51,52,53]. Furthermore, the shape of the horse’s head is much more affected by the changes in the color of the horse’s coat (compared with humans), potentially including patches of various colors, such as the commonly observed white star mark on the head [18]. In addition, extracting face shape boundaries from a horse face image is not easy, unlike a simple human face shape [18]. Most approaches for human face analysis assume a scaled image of equal width and height, which is not possible with a horse face unless the face is heavily distorted for larger pose angles [18].

Therefore, deep learning, especially CNN [54], can be used to extract and learn multiple features for an appropriate facial expression recognition system [55,56]. The use of CNNs has resulted in impressive performance in a variety of visual recognition tasks [57,58,59]. Recent research has demonstrated that despite being trained on image level labels, CNNs have a remarkable ability to localize objects [60,61,62,63]. In addition, it has been reported that CNNs can generalize this ability beyond localizing objects to start identifying exactly that region of the image being identified using the correct architecture [46]. Nevertheless, for facial expressions, most signals come from some regions of the face, such as the mouth and eyes, while others, such as ears and hair, play little role in the output [64]. This means that the machine learning framework should only focus on important parts of the face and be less sensitive to other face regions [39]. Therefore, in this study, important parts of the horse’s face were selected (the eyes, nose, and ears), and other parts were masked in black, as they could affect the analysis.

The reasons for choosing the eyes, nose, and ears as the main points of the horse’s face in our study were as follows. First, in the eyes, the position, expression, tension of the muscles in the dorsal and tail of the eyes, and whether the eyes are open or partially or completely closed were important observations [65]. In particular, a fixed gaze was previously considered a common pain symptom [16,20]. Orbital constriction with partially closed eyes was described as a symptom of pain after castration in horses by Della Costa et al. [37]. However, other researchers have suggested that this appearance in standing horses indicates dozing and represents a component of fatigue as a result of a surgical stress response following castration [16]. In addition, closing of eyes may reflect learned helplessness in response to chronic pain [65].

Second, wide-open nostrils with angled edges showed a weak association with lameness. In a previous study, medial dilatation of angled nostrils rather than round ones was associated with pain in stationary horses [16,37,66]. In case of riding horses, the interpretation can be confusing due to the breathing effort, depending on the horse’s fitness and the tasks to be performed [65].

Finally, the position of the ear is an important factor in differentiating claudication from healthy horses and was widely used in horse pain assessment in previous studies [13,16,37,66]. A flat ear was considered a very reliable pain indicator by observers evaluating horses with spondylitis [66]. Dyson et al. [65] also reported opposite ear positions, with one ear forward and the other backward or one ear to the side and the other to the back, as well as “behind the ear” as potential indicators of claudication. However, in our study, both ears could not be observed simultaneously in the profile view.

Therefore, according to the CAM results of this study, the weights are most concentrated on the eyes; thus, it is highly likely to have the greatest influence on the classification results. This study suggests that the pain expression classification significantly reflects the shape of the eye and the muscles around the eye.

Another important problem is that there are no studies reporting the existence of various facial expressions other than pain in horses. Dyson et al. [65] reported that the expressions of horses with pain and healthy horses due to exercise were described as an etogram (FEReq) for the expression of riding horses. In this study, it was determined whether horse facial expressions could be applied to a pain scoring system as an etogram, and it was found that the head position, ears, eyes, and nostrils contributed significantly, whereas beat position, lips, snout, and tongue contributed less [65]. However, there are no studies comparing the resting and post-exercise expressions of healthy horses. In addition, these etogram functions need to be collectively considered with pain scores rather than individual observations, thereby requiring automation. Moreover, the discomfort and pain felt by animals may be misunderstood.

In addition, some horses had already experienced hoof disease. Shin et al. [67] recently reported the prevalence of hoof disorders (thrush 4.2%, superficial hoof wall cracks 1.2%, white line disease 1.0%, hoof wall separation 0.6%, hoof wall defect 0.5%, laminitis 0.3%, and wounds 0.2%) in racing and riding horses in South Korea. Therefore, it is considered that there must have been many horses that experienced tension during horseshoeing due to pain from hoof disease when wearing horseshoes or the experience of touching the sensory part with the wrong nail of the horseshoeing. This is clearly different from the expression of pain shown in this study, and the expression of horses during horseshoeing was regarded as a tense expression.

As suggested by Anderson et al. [17], various pain faces may also exist in horses. We agree that it can contribute to more diverse and complex truths about pain expression compared to those currently observed on the horse facial expression pain scale. In this study, we discovered that not only pain but also various expressions exist in horses and developed a model for automatically recognizing and classifying horse facial expressions using deep learning. However, image-based research that can use video to select unbiased horse facial expressions should be conducted. In addition, automatic recognition of pain and stress can greatly enhance the identification of pain and other emotional states. Therefore, the study described here could be applied to other fields of research, providing new information about the animal’s point of view. 

## 6. Conclusions

Pain assessment monitoring and animal welfare state evaluation are essential. In our study, we developed an equine facial expression recognition model to automatically analyze these factors using deep learning by classifying them into four labels. Therefore, recognition of stress and other emotional states based on facial expressions can aid in presenting objective indicators by improving the quality of equine welfare.

## Figures and Tables

**Figure 1 vetsci-10-00283-f001:**
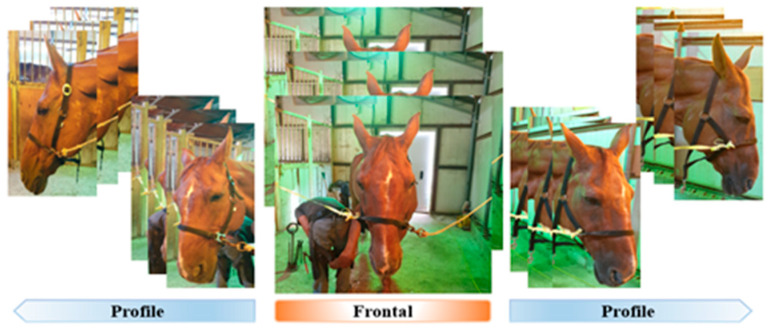
Images of a horse’s face from various angles. Each horse was photographed three times from five directions (left, between left and front, front, between front and right, and right), resulting in a total of 15 images per horse.

**Figure 2 vetsci-10-00283-f002:**
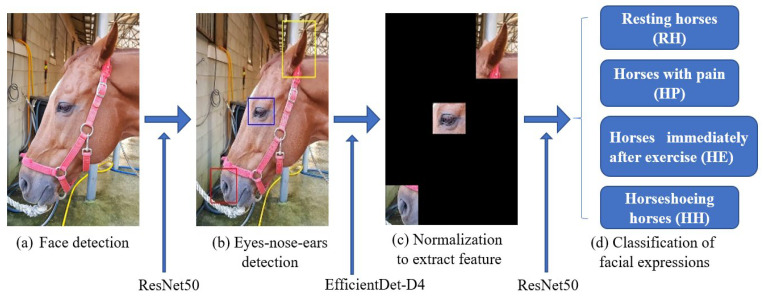
Pipeline of our automatic approach for the classification of various facial expressions of horses. (**a**) face detection, (**b**) eyes-nose-ears detection, (**c**) normalization to extract feature, (**d**) classification of facial expressions.

**Figure 3 vetsci-10-00283-f003:**
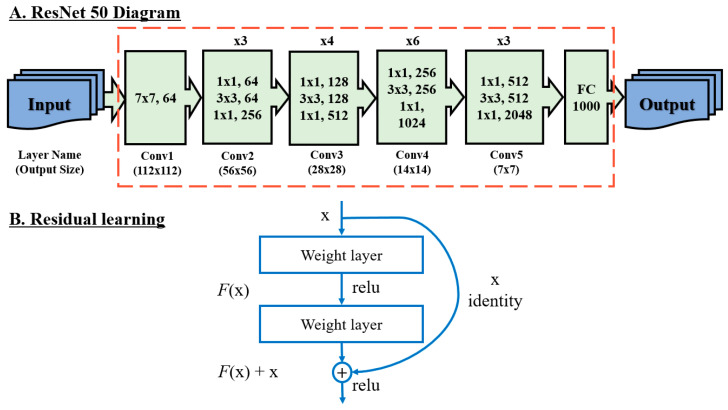
Architecture of residual network (ResNet) 50 and a building block for residual learning. (**A**) ResNet 50 diagram, (**B**) residual learning.

**Figure 4 vetsci-10-00283-f004:**
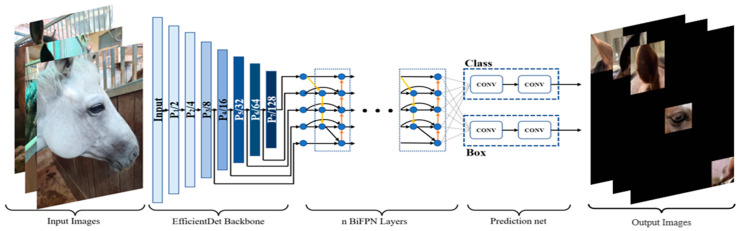
Architecture and performance comparison of EfficientDet. It becomes heavier from D1 to D7; D4 was selected, known as the appropriate level.

**Figure 5 vetsci-10-00283-f005:**
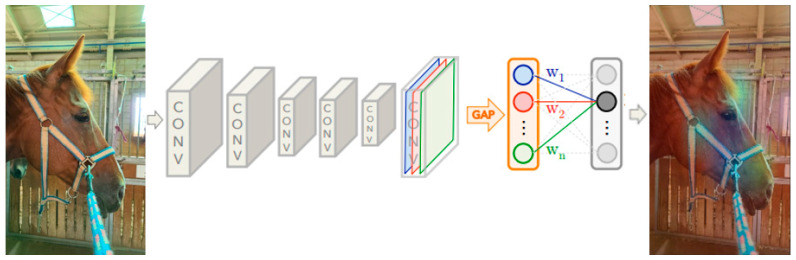
Architecture of class activation mapping; the predicted class scores are remapped to previous convolution layers to generate class activation maps (CAMs). The CAM highlights the class-specific discriminative regions.

**Figure 6 vetsci-10-00283-f006:**
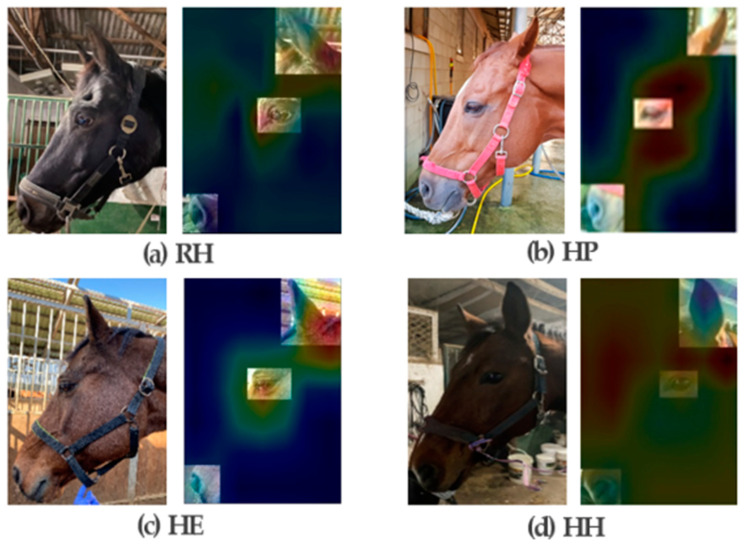
The maps highlight the discriminative image regions used for image classification. A simple modification of the global average pooling layer combined with our class activation mapping (CAM) technique allows the classification-trained CNN to both classify the image and localize class-specific image regions in a single forward pass: (**a**) RH; resting horses, (**b**) HP; horses with pain, (**c**) HE; horses immediately after exercise, and (**d**) HH; horseshoeing horses.

**Figure 7 vetsci-10-00283-f007:**
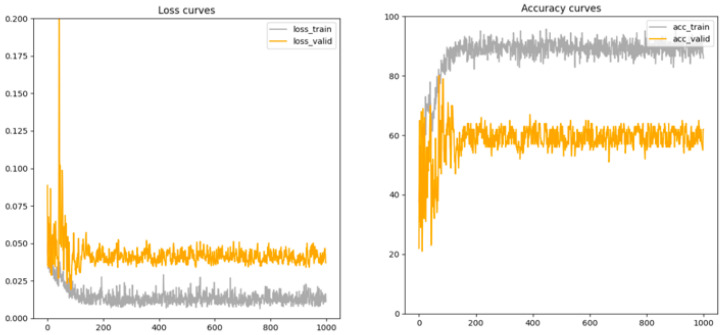
Accuracy and loss curves for the training and validation sets.

**Table 1 vetsci-10-00283-t001:** Confusion matrix for diagnostic model evaluation.

Model Classification Results	Actual Results	
	True	False
True	True positive (*TP*)	False positive (*FN*)
False	False negative (*FN*)	True negative (*TN*)

**Table 2 vetsci-10-00283-t002:** Comparison of the accuracy of the frontal and profile views for normalizing equine facial posture.

	N	Accuracy (%)
Frontal	162/166	97.59
Profile	181/187	99.45
Mean	343/353	98.52

**Table 3 vetsci-10-00283-t003:** Accuracy of equine facial keypoint detection classification.

	Training		Validation		Test	
	*n* = 339	Accuracy	*n* = 137	Accuracy	*n* = 273	Accuracy
Resting	80/80	100.0	21/25	84.0	97/105	92.38
Feeling pain	77/78	98.72	20/27	74.07	47/58	77.41
Exercising	78/81	96.29	19/25	76.0	42/50	84.0
Horseshoeing	100/100	100.0	55/60	91.67	57/60	95.0
Total		98.75		81.44		88.1

**Table 4 vetsci-10-00283-t004:** Accuracy according to the classification of equine pain in the test.

Classification	*n* = 58	Accuracy (%)
Paper	7/7	100
Google *	9/12	75
Surgery	17/19	89.47
Laminitis	6/6	100
Fattened horses	8/8	100
Severe lameness	0/6	0

* Photos taken from Google were selected as photos in the article.

## Data Availability

The data presented in this study are available on request from the corresponding author.

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
