# Peer review of "Analysis of Various Facial Expressions of Horses as a Welfare Indicator Using Deep Learning"

_vetsci, 2023, doi:10.3390/vetsci10040283_

Round 1
Reviewer 1 Report
Overall, an interesting study looking at assessing various expressions in horses in multiple situations using machine learning. There are some consistency errors throughout the manuscript, especially with the 4 classifications which were used to assess the equine facial expression. Eg. In the simple summary ‘comfort, tension, excitation’, in the methods ‘resting, pain, exercising and horseshoeing’, also, ‘break’ and ‘normal’ were used interchangeably. It would aid readability considerably if these situations (rather than emotive states) were used consistently throughout the paper, with attention being paid to the fact that these were situations being identified, rather than emotive states.
The use of ‘we’ throughout the paper could be adjusted so the paper is written in third person as is standard for scientific literature. Please define all abbreviations at first use. The sections of the paper don’t fit into the headings of Introduction, Methods, Results, Discussion and Conclusions – some restructuring of the information into the appropriate sections would aid in coherence and readability.
Were there any ethical considerations/approval sought for the study?
Simple Summary
There is a heavy weighting on pain expression detection, whereas the main findings appear to show that the programme can discriminate between horses in different situations. It might be better to place less stress on pain identification and provide a more balanced summary of your results. Also it would be helpful to explicitly state that the model is a computer model using machine learning.
Line 13 – The most recent literature has a general consensus that horses do feel pain, as explained later in the introduction, so I’m not sure why it says there is ‘little consensus’ here.
Line 14 – What do you mean by beneficial pain? As this isn’t mentioned elsewhere in the paper, consider rewording or removing.
Line 18 – define your 4 labels – and please keep consistency of labelling throughout the paper
Abstract
Again, there is a large emphasis on the pain expression, when this was only one part of the model. I would suggest amending to align with the aims and results of your study.
Line 26 – ‘feeling pain’ – I would suggest that we cannot know for sure what the horse is feeling, so perhaps reword to reflect that the horse is in a state where they would be expected to experience pain.
Line 31 – did your model actually assess different expressions according to degree and type of pain? These are interesting statements, but consider rewording to clarify that these are possibilities, but not what your results show.
Introduction
Good and interesting search of the literature.
Line 77-79. Unsure what this sentence means, I think it may be missing a word at the end.
Line 86-87. What limitations? This sentence seems out of place.
Line 100: Please define your terms at first use e.g CNN.
Line 104-6: Inconsistency in category labels (comfort, pain etc).
Line 106 : is the programme simply a welfare indicator?, consider rewording.
Materials and Methods
This entire section contains background information which would be better placed in the introduction section. Consider condensing this entire section and integrating it into the Introduction, perhaps with the use of subheading to aid reading flow.
Line 148: Missing “The” (Majority of modern machine…)
Lines 152-170: Consider condensing this information and relating to the equine application
Lines 165-166: “….making the horses and humans face shapes.” This sentence doesn’t make sense, please rewrite.
Line 172-173: Please make clear distinctions about what species you are considering in each of your examples, and their relevance to your current study
Results
The first part of your results should be placed in your methods section – description of animals used image collection, methodology, class activation mapping and model convergence. The results section should begin with your RESULTS from the use of the model.
Please clarify your use of still images, rather than video – these terms are used interchangeably and it is confusing to determine which was actually used in your study.
Use of the term ‘exercising’ – as the pictures were taken after an exercise session, this descriptor is not entirely accurate.
Please quantify the number of images you used for each classification
Lines 208-211: Were the pictures taken at the vet hospital? Before or after procedures?
Line 213: What does ‘hooves turned’ mean? Was the image obtained taken in their stable or under what conditions?
Line 214-5: more detail required here – how many images were used, what papers were they extracted from? Table 4 mentions Google, but this is not mentioned here?. Integrate lines 402-8, and state how many images were sourced from where and what the selection criteria were.
Lines 216-219: Did all horses follow the same exercise procedure?
Line 221-225: ‘groomed for hooves’ – please use correct terminology. Were the pictures taken before, after or during the farriery treatment?
Line 242: suggest ‘tested’ instead of proposed
Line 243-4: How many images were used in each subset?
Line 251: break – consistency of terms – figure 2 uses ‘normal’ and resting has been used elsewhere
Line 255: is PyTorch defined/explained?
Line 289: define HGS
Line 293-295: This is repeating methods already described – try to condense and collect together methods sections
Lines 309-314: Better suited in the discussion
Line 357: use of ‘break’ rather than resting or normal or stable
Line 371: Results section should start here. Line 377 mentions human expressions – this is not part of your results, and again may be better suited to the discussion.
Figure 6: Consistency of terms (Break, resting)
Line 401: State what the accuracy of pain classification was – low relative to what?
Lines 410-416: This paragraph needs more explanation, and is better suited to the discussion than results. Were the researches filming (using video) or taking static pictures as stated in your methods?. Perhaps introduce the difference between acute and chronic pain expressions?
Table 4: Would be helpful if references were added here for the source of the images. How were the pictures sourced from Google assessed for relevance/accuracy as to the horses situation?
Figure 7: No axis labels
Discussion
As suggested above, some of the information from results (and methods sections) might be better suited to being placed in the discussion.
Line 444: Replace fur with ‘hair’ or ‘coat’
Line 445: spelling ‘patches’
Line 446: ‘bounding box’ – presumably you mean for the computer images, but perhaps reword for clarity
Line 489: what is ‘beat position’. Maybe replace ‘snout’ with nose or muzzle.
Line 494: Which horses had experienced hoof disease? In this study? It seems that you are proposing that horses may be tense during horseshoeing. This paragraph is difficult to read, and requires rewording. Line 499 ‘ferries’ is misspelt and appears to be conjecture – do you have any references for this (though it is a possibility), I think it needs to be clear that this is the authors suggestion, and not a proven outcome.
Line 512: be careful with your assumptions, we do not know the horses ‘point of view’ though we can classify the expressions of the horse as they relate to the different situations you have proposed. With more research this may be able to be extended to assess if similar expressions are held by the horse in situations where we would expect the same emotions to be felt as have been quantified in this study.
Conclusions
Line 516: You have not assessed emotions, but the expressions in diverse situations and then related to them to the likely emotions. Be careful with the terminology. Suggest rewording.
Author Response
Reviewer 1
Q1. Overall, an interesting study looking at assessing various expressions in horses in multiple situations using machine learning. There are some consistency errors throughout the manuscript, especially with the 4 classifications which were used to assess the equine facial expression. Eg. In the simple summary ‘comfort, tension, excitation’, in the methods ‘resting, pain, exercising and horseshoeing’, also, ‘break’ and ‘normal’ were used interchangeably. It would aid readability considerably if these situations (rather than emotive states) were used consistently throughout the paper, with attention being paid to the fact that these were situations being identified, rather than emotive states.
The use of ‘we’ throughout the paper could be adjusted so the paper is written in third person as is standard for scientific literature. Please define all abbreviations at first use. The sections of the paper don’t fit into the headings of Introduction, Methods, Results, Discussion and Conclusions – some restructuring of the information into the appropriate sections would aid in coherence and readability.
Were there any ethical considerations/approval sought for the study?
A. I took your comments and made the best corrections.
Q2. There is a heavy weighting on pain expression detection, whereas the main findings appear to show that the programme can discriminate between horses in different situations. It might be better to place less stress on pain identification and provide a more balanced summary of your results. Also it would be helpful to explicitly state that the model is a computer model using machine learning.
Line 13 – The most recent literature has a general consensus that horses do feel pain, as explained later in the introduction, so I’m not sure why it says there is ‘little consensus’ here.
Line 14 – What do you mean by beneficial pain? As this isn’t mentioned elsewhere in the paper, consider rewording or removing.
- As suggested by the reviewer, I have revised the line 13 and 14 of the section Simple summary.
“However, there was no study that measured the level of pain, whether acute pain and chronic pain could be distinguished, or could be mistaken for pain.”
Line 18 – define your 4 labels – and please keep consistency of labelling throughout the paper.
- As suggested by the reviewer, I have revised the section Simple summary.
“We captured not only pain expressions but also comfort, tension, and excitation expressions as images by classifying them into four labels resting horses (RH), horses with pain (HP), horses immediately after exercise (HE) and horseshoeing horses (HH).”
Q3. Again, there is a large emphasis on the pain expression, when this was only one part of the model. I would suggest amending to align with the aims and results of your study.
Line 26 – ‘feeling pain’ – I would suggest that we cannot know for sure what the horse is feeling, so perhaps reword to reflect that the horse is in a state where they would be expected to experience pain.
- As suggested by the reviewer, I have revised “horse with pain”.
Line 31 – did your model actually assess different expressions according to degree and type of pain? These are interesting statements, but consider rewording to clarify that these are possibilities, but not what your results show.
- As suggested by the reviewer, I have revised Abstract.
“These results imply that various facial expressions in addition to pain may exist in horses depending on the situation, degree of pain, and type of pain experienced by horses.”
Q4. Introduction.
Line 77-79. Unsure what this sentence means, I think it may be missing a word at the end.
- As suggested by the reviewer, I have revised paragraph.
“In recent decades, some tools have been developed primarily for the diagnosis and evaluation of horse pain in certain clinical situations [9,13,28,29]. However, manually scored pain assessment practices, performed repeatedly by observers, may increase the risk of non-compliance, particularly [30]. Despite the validity gained for the distinction between healthy and pain-experienced horses, the observer's ability to recognize the normal behavior of horse can have inconsistent results [30]. Thus, there is an increased risk of underestimating pain or misinterpreting behavior in the context of assessment of well-being and quality of life [30].”
Line 86-87. What limitations? This sentence seems out of place.
- As suggested by the reviewer, deleted the sentence as it is unnecessary.
Line 100: Please define your terms at first use e.g CNN.
- As suggested by the reviewer, CNN’s full text was inserted.
“Convolutional Neural Networks (CNN)”
Line 104-6: Inconsistency in category labels (comfort, pain etc).
- As suggested by the reviewer, category labels have been unified.
“resting horses (RH), horses with pain (HP), horses immediately after exercise (HE) and horseshoeing horses (HH)”
Line 106 : is the programme simply a welfare indicator?, consider rewording.
- As suggested by the reviewer, I have revised paragraph.
“Therefore, the purpose of this study is to present more objective indicators for animal welfare by analyzing various facial expressions of horses.”
Q5. Materials and Methods
This entire section contains background information which would be better placed in the introduction section. Consider condensing this entire section and integrating it into the Introduction, perhaps with the use of subheading to aid reading flow.
Line 148: Missing “The” (Majority of modern machine…)
Lines 152-170: Consider condensing this information and relating to the equine application
Lines 165-166: “….making the horses and humans face shapes.” This sentence doesn’t make sense, please rewrite.
Line 172-173: Please make clear distinctions about what species you are considering in each of your examples, and their relevance to your current study
- As suggested by the reviewer, I incorporated some into the introduction for reading flow, but deleted this section because it contained too much information.
Q6. Results
The first part of your results should be placed in your methods section – description of animals used image collection, methodology, class activation mapping and model convergence. The results section should begin with your RESULTS from the use of the model.
- As suggested by the reviewer, I corrected as Materials and Methods.
Please clarify your use of still images, rather than video – these terms are used interchangeably and it is confusing to determine which was actually used in your study.
Use of the term ‘exercising’ – as the pictures were taken after an exercise session, this descriptor is not entirely accurate.
- As suggested by the reviewer, I corrected as “horses immediately after exercise”.
Please quantify the number of images you used for each classification.
- As suggested by the reviewer, the number of images was inserted.
Lines 208-211: Were the pictures taken at the vet hospital? Before or after procedures?
- All horses with pain except fattening horses were photographed at the equine veterinary hospital of the KRA.
Line 213: What does ‘hooves turned’ mean? Was the image obtained taken in their stable or under what conditions?
- It is a horse that has been raised in a state where it is not managed, and refers to hoof deformity.
Line 214-5: more detail required here – how many images were used, what papers were they extracted from? Table 4 mentions Google, but this is not mentioned here?. Integrate lines 402-8, and state how many images were sourced from where and what the selection criteria were.
- As suggested by the reviewer, I corrected and the number of images was inserted.
Lines 216-219: Did all horses follow the same exercise procedure?
- Yes. Most exercise procedures were similar.
Line 221-225: ‘groomed for hooves’ – please use correct terminology. Were the pictures taken before, after or during the farriery treatment?
- As suggested by the reviewer, I corrected as farriery treatment. And the pictures were taken during the farriery treatment.
Line 242: suggest ‘tested’ instead of proposed
- As suggested by the reviewer, I corrected.
Line 243-4: How many images were used in each subset?
- As suggested by the reviewer, I corrected.
“We trained the model on a subset of that dataset (N=339), validated it on a validation set (N=137), and reported the accuracy over the test set (N=273).”
Line 251: break – consistency of terms – figure 2 uses ‘normal’ and resting has been used elsewhere
- As suggested by the reviewer, I’m unified everywhere. And I corrected Figure 2.
Line 255: is PyTorch defined/explained?
- As suggested by the reviewer, I inserted.
Line 289: define HGS
- As suggested by the reviewer, I inserted.
Line 293-295: This is repeating methods already described – try to condense and collect together methods sections
- As suggested by the reviewer, I corrected.
Lines 309-314: Better suited in the discussion
- As suggested by the reviewer, the sentences moved to discussion.
Line 357: use of ‘break’ rather than resting or normal or stable
- I corrected about four labels.
Line 371: Results section should start here. Line 377 mentions human expressions – this is not part of your results, and again may be better suited to the discussion.
- As suggested by the reviewer, the sentences moved to discussion.
Figure 6: Consistency of terms (Break, resting)
- As suggested by the reviewer, I’m unified everywhere.
Line 401: State what the accuracy of pain classification was – low relative to what?
- As suggested by the reviewer, I corrected.
“Overall, the average classification accuracy of the four labels was high, but the accuracy of pain classification was the lowest.”
Lines 410-416: This paragraph needs more explanation, and is better suited to the discussion than results. Were the researches filming (using video) or taking static pictures as stated in your methods?. Perhaps introduce the difference between acute and chronic pain expressions?
- As suggested by the reviewer, the sentences moved to discussion.
Table 4: Would be helpful if references were added here for the source of the images. How were the pictures sourced from Google assessed for relevance/accuracy as to the horses situation?
- As suggested by the reviewer, I inserted in Table 4.
“Photos taken from Google were selected as photos in the article.”
Figure 7: No axis labels
Q7. Discussion
As suggested above, some of the information from results (and methods sections) might be better suited to being placed in the discussion.
Line 444: Replace fur with ‘hair’ or ‘coat’
- As suggested by the reviewer, I corrected as “coat”.
Line 445: spelling ‘patches’
- As suggested by the reviewer, I corrected.
Line 446: ‘bounding box’ – presumably you mean for the computer images, but perhaps reword for clarity
A. As suggested by the reviewer, I corrected.
“In addition, extracting face shape boundaries from a horse face image is not easy, unlike a simple human face shape.”
Line 489: what is ‘beat position’. Maybe replace ‘snout’ with nose or muzzle.
- The beat position is the lip position affected by the biting of the horse.
Line 494: Which horses had experienced hoof disease? In this study? It seems that you are proposing that horses may be tense during horseshoeing. This paragraph is difficult to read, and requires rewording. Line 499 ‘ferries’ is misspelt and appears to be conjecture – do you have any references for this (though it is a possibility), I think it needs to be clear that this is the authors suggestion, and not a proven outcome.
A. As suggested by the reviewer, I corrected.
“Therefore, it is considered that there must have been many horses that experienced tension during horseshoeing due to pain from hoof disease when wearing horseshoes or the experience of touching the sensory part with the wrong nail of the horseshoeing.”
Line 512: be careful with your assumptions, we do not know the horses ‘point of view’ though we can classify the expressions of the horse as they relate to the different situations you have proposed. With more research this may be able to be extended to assess if similar expressions are held by the horse in situations where we would expect the same emotions to be felt as have been quantified in this study.
- As suggested by the reviewer, I agreed.
Q8. Conclusions
Line 516: You have not assessed emotions, but the expressions in diverse situations and then related to them to the likely emotions. Be careful with the terminology. Suggest rewording.
- Unnecessary sentences have been deleted.

Reviewer 2 Report
The paper is very interesting and reviews the topic of pain recognition in horses. The authors also mention that pain might be difficult to detect since it is not expressed all the time and thus it could be that at the time the photographs are taken the horse did not express pain.
My main concern about the paper is that includes quite detailed explanations about the various deep learning networks that are used in the experiments. I personally do not think that readers of this journal will gain any insight from reading these explanations. A more high level description should be given, such as the input/output of the networks. For people who are versed with these networks these details are known, or not important.
In addition, what is missing is a confusion matrix. Not a 2x2 matrix but a 4x4 in which the confusion between the classes and an explanation on between which classes the algorithm makes mistakes and maybe a suggested explanation.
Author Response
Reviewer 2
Q1. The paper is very interesting and reviews the topic of pain recognition in horses. The authors also mention that pain might be difficult to detect since it is not expressed all the time and thus it could be that at the time the photographs are taken the horse did not express pain.
My main concern about the paper is that includes quite detailed explanations about the various deep learning networks that are used in the experiments. I personally do not think that readers of this journal will gain any insight from reading these explanations. A more high level description should be given, such as the input/output of the networks. For people who are versed with these networks these details are known, or not important.
In addition, what is missing is a confusion matrix. Not a 2x2 matrix but a 4x4 in which the confusion between the classes and an explanation on between which classes the algorithm makes mistakes and maybe a suggested explanation.
- First, thank you for reading and reviewing my thesis with interest.
Since the inputs/outputs of the network are calculated automatically and the resulting values come out, a higher level explanation was not possible.
Also, as suggested by the reviewer, I corrected as 4x4 matrix.
